# Risk Factors for Hiccups after Deep Brain Stimulation of Subthalamic Nucleus for Parkinson’s Disease

**DOI:** 10.3390/brainsci12111447

**Published:** 2022-10-26

**Authors:** Bin Wu, Yuting Ling, Changming Zhang, Yi Liu, Ruoheng Xuan, Jiakun Xu, Yongfu Li, Qianqian Guo, Simin Wang, Lige Liu, Lulu Jiang, Zihuan Huang, Jianping Chu, Ling Chen, Nan Jiang, Jinlong Liu

**Affiliations:** 1Department of Neurosurgery, The First Affiliated Hospital of Sun Yat-sen University, Guangzhou 510080, China; 2Department of Neurology, The First Affiliated Hospital of Sun Yat-sen University, Guangzhou 510080, China; 3Department of Anesthesiology, The First Affiliated Hospital of Sun Yat-sen University, Guangzhou 510080, China; 4Department of Radiology, The First Affiliated Hospital of Sun Yat-sen University, Guangzhou 510080, China

**Keywords:** hiccups, Parkinson’s disease, deep brain stimulation, subthalamic nucleus, risk factors

## Abstract

Background: After deep brain stimulation (DBS), hiccups as a complication may lead to extreme fatigue, sleep deprivation, or affected prognosis. Currently, the causes and risk factors of postoperative hiccups are unclear. In this study, we investigated the risk factors for hiccups after DBS of the subthalamic nucleus (STN) for Parkinson’s disease (PD) under general anesthesia. Methods: We retrospectively included patients who underwent STN DBS in the study, and collected data of demographic characteristics, clinical evaluations, and medications. According to the occurrence of hiccups within seven days after operation, the patients were divided into a hiccups group and non-hiccups group. The potentially involved risk factors for postoperative hiccups were statistically analyzed by logistic regression analysis. Results: A total of 191 patients were included in the study, of which 34 (17.80%) had postoperative transient persistent hiccups. Binary univariate logistic regression analysis showed that male, higher body mass index (BMI), smoker, Hoehn and Yahr stage (off), preoperative use of amantadine, hypnotic, Hamilton anxiety scale and Hamilton depression scale scores, and postoperative limited noninfectious peri-electrode edema in deep white matter were suspected risk factors for postoperative hiccups (*p* < 0.1). In binary multivariate logistic regression analysis, male (compared to female, OR 14.00; 95% CI, 1.74–112.43), postoperative limited noninfectious peri-electrode edema in deep white matter (OR, 7.63; 95% CI, 1.37–42.37), preoperative use of amantadine (OR, 3.64; 95% CI, 1.08–12.28), and higher BMI (OR, 3.50; 95% CI, 1.46–8.36) were independent risk factors for postoperative hiccups. Conclusions: This study is the first report about the risk factors of hiccups after STN DBS under general anesthesia for PD patients. The study suggests that male, higher BMI, preoperative use of amantadine, and postoperative limited noninfectious peri-electrode edema in deep white matter are independent risk factors for postoperative hiccups of STN-DBS for PD patients. Most hiccups after STN-DBS for PD patients were transient and self-limiting.

## 1. Introduction

Hiccups refer to the sudden, involuntary, and spasmodic contraction of the diaphragm and external intercostal muscles, resulting in rapid inhalation, glottic closure, and a “HIC” sound [1,2,3,4]. Hiccups are more common in gastrointestinal diseases or central nervous system diseases. It was reported that 20% of patients with Parkinson’s disease (PD) and 10% of patients with esophageal reflux symptoms complain of repeated hiccups [1]. The classification of hiccups’ severity is usually defined by their duration: acute (<48 h), persistent (48 h to 1 month), and intractable (>1 month) [5,6]. Deep brain stimulation (DBS) related hiccups are mostly transient and persistent (<14 days) [7]. Although the hiccups of most PD patients are self-limiting and not life-threatening, continuous hiccups affect these patients’ quality of life and emotions by interfering with their diet, language communication, and sleep. Hiccups can lead to extreme fatigue, sleep deprivation, weight loss, depression, anxiety, aspiration pneumonia, and diaphragm rupture [8,9,10]. Moreover, even relatively short episodes of hiccups may have serious consequences for patients.

The pathophysiological mechanism of hiccups has been proposed to be associated with diaphragm dysfunction [1]. Currently, it is generally believed that hiccups are caused by a “reflex arc” with afferent, central, and efferent components [11,12]. Afferent components include the vagus nerve, the phrenic nerve, and sympathetic nerve fibers (T6–T12). The parts of the central nervous system involved in hiccup response include the superior spinal cord (C3-C5), the medullary brainstem near the respiratory center, the reticular structure, and the hypothalamus. Dopaminergic and γ- Aminobutyric acid (GABAergic) neurotransmitters can also regulate this central mechanism. Efferent components include the phrenic nerve, the diaphragm, the accessory nerve, the intercostal muscle, the vagus nerve, the recurrent laryngeal branch, and the glottis. Therefore, the whole reflection arc involves many components, and the process is lengthy. However, any process that affects the reflex arc afferent, central or efferent components may trigger hiccups. Therefore, it is a challenge to study the risk factors of hiccups.

Common inducing factors include functional and organic, and peripheral and central factors [1]. Gender is a common significant risk factor, and men are more at risk [8,12,13]. Postoperative persistent hiccups are related to gastric dilatation, phrenic nerve stimulation, and vagal surgery [14,15,16,17]. In addition, organic lesions involving the brain stem, especially those involving the medulla oblongata, such as multiple sclerosis, stroke, and tumors, have also been reported to be related to intractable hiccups [18,19,20,21,22]. For PD patients, the incidence of hiccups is 20%, which is higher than that of other central diseases, and is related to the disease itself or dopaminergic drugs [13,18,23]. Some studies believe that hiccups may precede the motor symptoms of PD, and that it is more likely to be a type of PD-associated motor disorder symptom [18,24]. Some studies suggest that hiccups are associated with antiparkinsonian medications, especially dopamine agonists [25,26,27,28,29,30]. Additionally, some studies have reported that post-DBS hiccups may be related to noninfectious edema around the electrode [7]. However, due to the limited number of studies on hiccups, the factors affecting post-DBS hiccups remain to be explored. We retrospectively collected data on the demographic characteristics, clinical evaluations, and medications of PD patients receiving DBS of the subthalamic nucleus (STN) to determine the risk factors of postoperative hiccups.

## 2. Materials and Methods

### 2.1. Subjects

This is a retrospective case-control observational research. The purpose of this study is to identify the risk factors of hiccups after STN DBS for PD patients under general anesthesia. A total of 248 PD patients who underwent bilateral STN DBS under general anesthesia in the First Affiliated Hospital of Sun Yat-sen University from April 2016 to December 2021 was identified. Hence, 57 patients were excluded from the analysis, among which two of them underwent other brain surgery, 34 patients lacked preoperative motor and nonmotor evaluation results, and 21 patients did not have postoperative magnetic resonance imaging (MRI) data. Therefore, 191 cases were included in this study (Figure 1). The patients were divided into two groups according to whether there had hiccups within 7 days after operation. There were 34 people in the hiccups group and 157 people in the non-hiccups group. The study protocol was approved by the local ethics committee (No. 2021-798) and was in line with the Declaration of Helsinki. Since this study was retrospective, we abandoned the requirement for informed consent.

### 2.2. Surgical Procedure

Preoperative localization and surgical procedures were similar to a previous study [31]. The brain was preoperatively scanned with 3T-MRI (signa excite; GE Medical Systems, Milwaukee, WI, USA) to identify and locate STN. The images were transmitted to the neuronavigational workstation through the image archiving and communication system. On the day of operation, the head frame was fixed to the patient’s skull under local anesthesia and the computed tomography (CT) scanning was performed. The CT image was fused with the preoperative MRI image to calculate the coordinates of STN and the planned electrode trajectory were obtained. The anesthesia process was performed by the same anesthesiologist using a unified scheme. For more details, refer to the corresponding part of the previous study [31] (sedative drugs: propofol or etomidate, analgesic drugs: remifentanil and sufentanil). During the operation, CIS atracurium was used to maintain muscle relaxation and ventilator support. Adjuvant medications included palonosetron, flurbiprofen, and dexmedetomidine. Microelectrode recording of bilateral STN and passive movements of the contralateral extremities were performed. After determining the best trajectory, the macroelectrode were implanted. Implantable pulse generator were performed under general anesthesia. All patients were transferred to ICU after operation. The day after the operation, CT of head was scanned to determine whether there was intracranial hemorrhage, and 1.5T-MRI was scanned 5–7 days postoperatively to determine the accuracy of the electrodes. The IPG was turned on one month after the surgery.

### 2.3. Data Collection

In this study, postoperative hiccup was defined as hiccup occurred within 7 days after STN-DBS surgery for PD patients. The occurrence of hiccups was extracted from the routine clinical records of doctors and nurses, and a specific record of the postoperative complications of DBS surgery. The records of hiccup were based on clinical observation of doctors and nurses, or reports from the patients themselves.

The clinical characteristics of PD included the duration of PD, age at onset, duration of antiparkinsonian medication, and disease severity (Hoehn and Yahr stage; off stage and on stage). Body mass index (BMI) was calculated and divided into three groups (<18.5, 18.5–23.9, >23.9).

The preoperative clinical evaluations included motor evaluations and nonmotor evaluations. Motor evaluation included the levodopa challenge test, motor classification, motor score (part 3 motor examination of the Movement Disorder Society-Unified Parkinson’s Disease Rating Scale, MDS-UPDRS III). Nonmotor evaluations involved cognition (Mini-Mental State Examination, MMSE), anxiety (Hamilton Anxiety Scale, HAMA), depression (Hamilton Depression Scale, HAMD), and dysarthria, and apathy and constipation.

The total number of microelectrodes used for recording trajectories and anesthesia time were obtained through operation and anesthesia electronic medical records. In particular, we analyzed whether there was limited edema of deep white matter around the electrode and extensive edema in the surface white matter on postoperative T2-MRI images and if they were associated with hemorrhage or infection. This analysis was performed according to the diagnostic method of peri-electrode edema after DBS suggested by Saitoh et al. [32].

Preoperative medication was also recorded, including levodopa equivalent daily dose (LEDD), antiparkinsonian medications (amantadine, trihexyphenidyl, pramipexole, piribedil, ropinirole, catechol-O-methyl transferase inhibitors, type-B monoamine oxidase inhibitors), antipsychotics, and hypnotics. The use of sedatives (propofol or etomidate) during the operation and whether dexamethasone was used after the operation were also collected.

### 2.4. Statistical Analysis

The SPSS (version 26, IBM Corp, Armonk, New York, USA) software was used for analysis. A descriptive analysis was used to summarize the baseline characteristics of PD patients. The quantitative data were expressed as mean ± standard deviation, and the qualitative data were expressed as number of cases and percentages. We included 36 variables regarding demographic characteristics, clinical evaluations, and medications for univariate analysis. Binary univariate logistic regression was applied to recognized candidate factors for multivariate analysis according to the criterion of *p* < 0.10. In the multivariate analysis, the variables with statistical significance were analyzed by binary multivariate logistic regression (stepwise method) to determine the independent risk factors of postoperative hiccups. Significance was recognized when *p* < 0.05.

## 3. Results

### 3.1. Population Characteristics

Baseline characteristics of the 191 cases are listed in Table 1. The average age was 59.90 ± 9.26 years old, and the average age of onset was 48.77 ± 11.76 years old. There were 118 (61.78%) men. The average BMI was 23.92 ± 14.74 kg/m^2^, the average duration of PD was 9.74 ± 3.95 years, and the average medication time was 8.76 ± 3.55 years. 

### 3.2. Characteristics of Hiccups

Among the 191 PD patients undergoing STN DBS under general anesthesia, 34 patients had postoperative hiccups, with an incidence of 17.80%. Only one patient had a history of hiccups, which was suspected to be due to oral dexamethasone. The other 33 patients had not seen similar hiccups in the past. Hiccups usually start 2–3 days after operation and stop 4–6 days after operation. There were no obvious causes. These occurred usually 0.5–2 h after taking antiparkinsonian drugs or eating. There seemed to be no effective method to relieve the hiccups. Neither drug treatment (antiparkinsonian drugs, metoclopramide, and chlorpromazine) nor nondrug treatment (drinking water or holding breath) was effective. Fortunately, hiccups were all self-limiting and did not cause obvious discomfort and complications. Only a few patients complained that hiccups even occurred during sleep.

### 3.3. Univariate and Multivariate Analysis Results

After univariate analysis, we found that in the initial 36 variables, there were nine factors with statistical differences between the two groups (*p* < 0.10, see Table 2). The nine variables were male, BMI, preoperative smoking, Hoehn and Yahr stage (off), preoperative use of amantadine and hypnotics, abnormal preoperative HAMA and HAMD scores, and postoperative limited noninfectious peri-electrode edema in deep white matter (see Figure 2).

The above nine factors that may have affected the occurrence of postoperative hiccups were used as independent variables for binary multivariate logistic regression analysis. The results showed that there were four independent risk factors for postoperative hiccups: male (compared to female, OR 14.00, 95% CI 1.74–112.43, *p* = 0.01), higher BMI (OR 3.50, 95% CI 1.46–8.36, *p* = 0.01), preoperative use of amantadine (OR 3.64, 95% CI 1.08–12.28, *p* = 0.04), and postoperative limited noninfectious peri-electrode edema in deep white matter (OR 7.63, 95% CI 1.37–42.37, *p* = 0.02). Based on these, being male appears to be the primary independent risk factor for hiccups.

## 4. Discussion

In this study, the incidence of postoperative hiccups of DBS was 17.80% and the factors inducing hiccups are complex and diverse. Therefore, it is a challenge to study the risk factors of postoperative hiccups. This study is the first to report the risk factors of hiccups after STN DBS under general anesthesia in PD. Our results suggest that male, higher BMI, preoperative use of amantadine, and postoperative limited noninfectious peri-electrode edema in the deep white matter are independent risk factors for postoperative hiccups.

Gender is the most common risk factor for hiccups. Our results show that male is the most important predictor of postoperative hiccups (OR 14, *p* = 0.01). The incidence of postoperative hiccups in male PD patients is 14 times higher than that in females. In healthy subjects, there is no gender difference in the frequency of common hiccups. Through a literature review, one study has proved the advantages of men in patients with hiccups, especially in patients with noncentral nervous system causes [33]. However, some studies that followed up Spanish patients with gastroesophageal reflux by telephone found that the incidence of hiccups was significantly higher in women than in men [34]. In PD population, there is an unclear relationship between gender and hiccups, and most studies focus on dopaminergic drugs [13]. The gender difference for the occurrence of hiccups remains unclear. For hiccups unrelated to the central nervous system, men’s susceptibility to hiccups may be due to the lower synaptic threshold, while the afferent or efferent nerves in the hiccup reflex arc are more likely to be excited [33]. For central nervous system related hiccups, this difference in occurrence is associated with the neuroendocrine mechanism of central hormone receptors. This may be related to the gender difference of steroid hormone receptors in the hippocampus and hypothalamus [34].

In this study, postoperative limited noninfectious peri-electrode edema in deep white matter, which is an independent risk factor for postoperative hiccups. Hiccups involve central nerves such as the glossopharyngeal nerve, vagus nerve, nucleus tractus solitarius, nucleus ambiguous, and phrenic nerve. However, the exact mechanism behind the central connection of the hiccup reflex arc is unclear. It has long been reported that PD patients are prone to transient hiccups after DBS, which mainly affect the pallidum compared with those of the STN [7,23]. It has been reported that DBS-related hiccups are related to the edema around the electrode, which supports the results of this study. Non-infectious peri-electrode edema after DBS is not uncommon, with an incidence of 3.2–6.3% [35,36]. This kind of edema is usually noninfectious and small. It usually occurs within 72 h. There is no manifestation of neurological deficit, making it easy to be clinically ignored. Edema is divided into extensive edema on the surface of white matter around the electrode and limited edema in the deep white matter. The reason for this may be associated with the microtrauma of the blood–brain barrier that was caused by mechanical stimulation from the intraoperative microelectrode or lead implantation, or the immune reaction to electrode materials. This study shows that there is a significant correlation between the limited peri-electrode edema in the deep white matter and postoperative hiccups. The edema was located around the STN and substantia nigra on T2-MRI, suggesting that the structure of the fiber connection projection around the STN and substantia nigra, is related to the trigger of hiccups. The transient and self-limiting nature of edema coincided with the duration and self-limiting characteristics of hiccups.

We reported for the first time that preoperative amantadine can induce postoperative hiccups in patients with PD. Previous studies have investigated the drugs associated with hiccups in PD patients [27,28,29]. Among the dopamine related drugs, the dopamine agonists, aripiprazole and levodopa, have been identified as hiccup-inducing drugs. Controversially, many studies have reported that amantadine may be able to treat persistent hiccups [37,38]. The controversy is similar to that of pramipexole, which is also believed to induce or treat hiccups [13,28]. Although amantadine and pramipexole have different mechanisms of action, the relationship between dopaminergic drugs and hiccups is still unclear. Some reviews have pointed out that hiccups may be mediated by central (GABA, dopamine, and serotonin) and peripheral (epinephrine, norepinephrine, acetylcholine, and histamine) neurotransmitters [39,40]. The mechanism of amantadine may involve the promotion of dopamine release from intact dopaminergic neurons in the striatum or the inhibition of its synaptic reuptake. It can directly stimulate dopamine receptors and has a weak anticholinergic effect. Therefore, amantadine may induce hiccups by affecting dopamine and acetylcholine.

In this study, we show that higher BMI can predict the increased risk of postoperative hiccups. Higher BMI increases the risk of postoperative hiccups. A study investigated the age, height, and weight characteristics of patients with hiccups and found that older age, higher height, and greater weight indicated higher incidence of hiccups [12]. 

The limitation of this study is in its retrospective nature. In fact, 11 patients with postoperative hiccups could not be included in the analysis due to the lack of clinical evaluation results and postoperative MRI data. The sample size was relative not large. However, this study is the first comprehensive and objective analysis of the risk factors of postoperative hiccups based on demographic characteristics, clinical evaluations, surgical data, and medication. A larger cohort and multicenter data remain needed to further explore and verify the current research results.

## 5. Conclusions

The study suggests that male, higher BMI, preoperative use of amantadine, and postoperative limited noninfectious peri-electrode edema in deep white matter are independent risk factors for postoperative hiccups of STN-DBS for PD patients. Most hiccups after STN-DBS for PD patients were transient and self-limiting.

## Figures and Tables

**Figure 1 brainsci-12-01447-f001:**
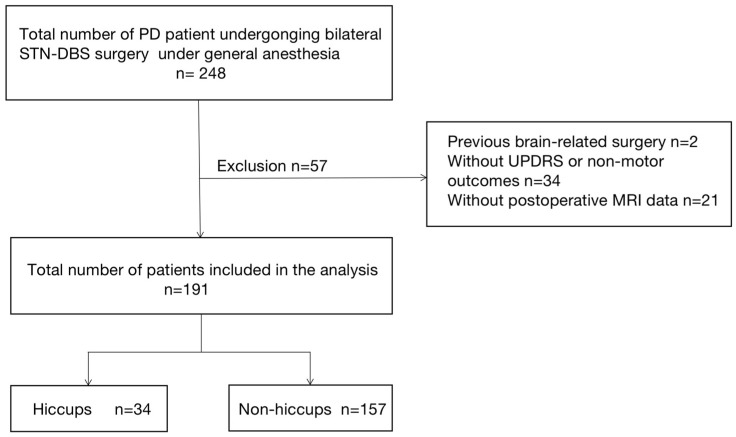
Data collection and analysis flowchart.

**Figure 2 brainsci-12-01447-f002:**
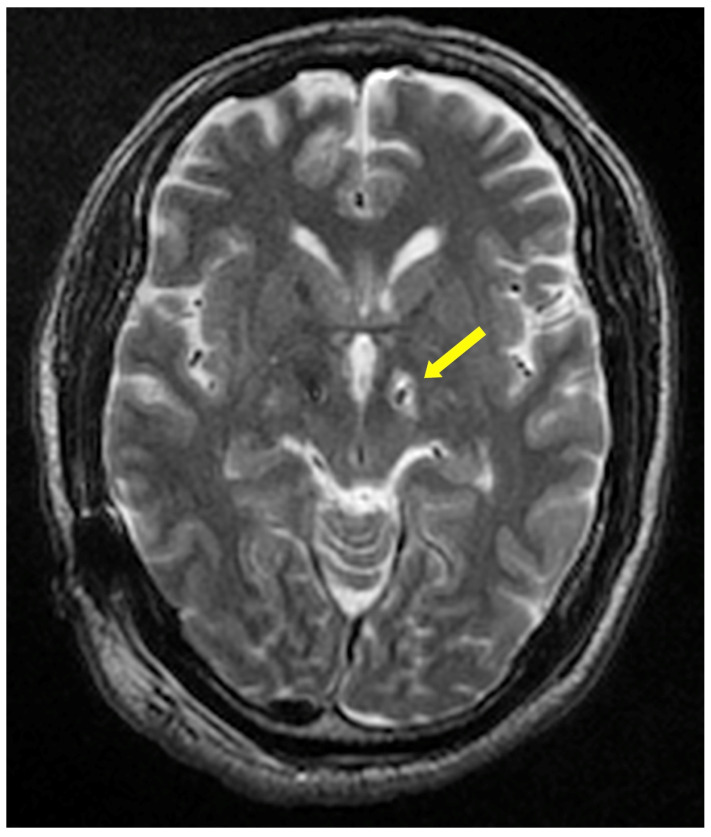
The yellow arrow indicates limited non-infectious peri-electrode edema in deep white matter on postoperative T2-MRI image.

**Table 1 brainsci-12-01447-t001:** Baseline characteristics and univariate analysis of postoperative hiccups in PD patients after STN-DBS under general anesthesia.

	Total	Non-Hiccups	Hiccups	*p* Value
	*n* = 191	*n* = 157	*n* = 34	
Demographic characteristics				
Age (years)	59.90 ± 9.26	60.29 ± 9.11	58.09 ± 9.86	0.21
Age at onset of PD (years)	48.77 ± 11.76	48.93 ± 11.68	48.03 ± 12.25	0.69
Male sex	118 (61.78)	85 (54.14)	33 (97.06)	<0.01
Body mass index (kg/m^2^)	23.92 ± 14.74	23.81 ± 16.22	24.44 ± 2.51	<0.01
Diabetes	15 (7.85)	12 (7.64)	3 (8.82)	0.82
Hypertension	47 (24.61)	38 (24.20)	9 (26.47)	0.78
Smoker (current/former)	25 (13.09)	17 (10.83)	8 (23.53)	0.05
Duration of PD (years)	9.79 ± 3.99	9.97 ± 4.15	8.97 ± 3.01	0.19
Drug duration (years)	8.76 ± 3.55	8.86 ± 3.62	8.30 ± 3.22	0.40
Clinical evaluations				
Hoehn and Yahr stage, off	3.186 ± 0.72	3.30 ± 0.73	2.99 ± 0.65	0.08
Hoehn and Yahr stage, on	2.178 ± 0.66	2.17 ± 0.66	2.24 ± 0.65	0.58
MDS-UPDRS III off	50.43 ± 16.28	50.82 ± 16.71	48.65 ± 14.21	0.48
Tremor domination	101 (52.88)	80 (50.96)	21 (61.76)	0.26
Levodopa challenge test (%)	61.70 ± 12.77	61.41 ± 13.06	63.02 ± 11.41	0.51
Dysarthria	149 (78.01)	120 (76.43)	29 (85.29)	0.26
Apathy	35 (18.32)	29 (18.47)	6 (17.65)	0.91
MMSE score	27.20 ± 3.16	27.05 ± 3.40	27.88 ± 1.57	0.17
HAMA score				0.01
<8, normal	121 (63.35)	93 (59.24)	28 (82.35)	
8–14, anxiety, may be	51 (26.70)	45 (28.66)	6 (17.65)	
15–21, anxiety (mild)	13 (6.81)	13 (8.28)	0	
>21, anxiety (moderate/severe)	6 (3.14)	6 (10.19)	0	
HAMD score				0.02
<8, normal	99 (51.83)	76 (48.41)	23 (67.65)	
8–20, suspicious depression	77 (40.31)	66 (42.04)	11 (32.35)	
>20, depression	15 (7.85)	15 (9.55)	0	
Constipation	143 (74.87)	116 (73.89)	27 (79.41)	0.50
Total number of MER trajectories	2.66 ± 1.18	2.66 ± 1.20	2.68 ± 1.07	0.64
Anesthesia time (minutes)	420.87 ± 63.92	422.64 ± 63.67	412.71 ± 65.38	0.41
Limited peri-electrode edema on T2	145 (75.92)	121 (70.07)	32 (94.12)	0.04
Extensive T2 hyperintensity in surface white matter around the electrode	153 (80.10)	121 (70.07)	24 (70.59)	0.42
Medications				
LEDD (mg)	818.85 ± 375.63	817.64 ± 399.83	824.43 ± 237.92	0.92
Amantadine	26 (13.61)	18 (11.46)	8 (23.53)	0.07
Trihexyphenidyl	24 (12.57)	17 (10.83)	7 (20.59)	0.13
Pramipexole	102 (52.88)	83 (52.87)	19 (55.88)	0.75
Piribedil	43 (22.51)	32 (20.38)	11 (32.35)	0.13
Ropinirole	11 (5.76)	9 (5.73)	2 (5.88)	0.97
COMT inhibitors	89 (46.60)	76 (48.41)	13 (38.24)	0.28
MAOB inhibitors	44 (23.04)	33 (21.02)	11 (32.35)	0.16
Preoperative antipsychotics	22 (11.52)	20 (12.74)	2 (5.88)	0.27
Preoperative hypnotics	64 (33.51)	59 (37.58)	5 (14.71)	0.02
Intraoperative use of etomidate	39 (20.42)	34 (21.66)	5 (14.71)	0.37
Postoperative use of dexamethasone	127 (66.49)	101 (64.33)	26 (76.47)	0.18

Numbers indicate means ± standard deviations (SD) or number (percentage). Differences between the non-hiccups group and hiccups group were analyzed using binary univariate logistic regression analysis. The candidate factors for further multivariate analysis were recognized according to the standard of *p* < 0.10. PD: Parkinson’s disease, STN: subthalamic nucleus, DBS: deep brain stimulation, MDS-UPDRS III: part 3 motor examination of the Movement Disorder Society-Unified Parkinson’s disease rating scale, MMSE: mini-mental state examination, HAMA: Hamilton anxiety scale, HAMD: Hamilton depression scale, MER: microelectrode recording, LEDD: Levodopa equivalent daily dose, COMT: catechol-O-methyl transferase, MAOB: type-B monoamine oxidase.

**Table 2 brainsci-12-01447-t002:** Univariate and multivariate analyses of postoperative hiccups in PD patients after STN-DBS under general anesthesia.

	Univariate Analysis				Multivariate Analysis			
	Odds Ratio	Lower	Upper	*p* Value	Odds Ratio	Lower	Upper	*p* Value
Male	27.95	3.73	209.47	<0.01	14.00	1.74	112.43	0.01
Body mass index	3.60	1.75	7.40	<0.01	3.50	1.46	8.36	0.01
Smoker (current/former)	2.53	0.99	6.48	0.05	-	-	-	-
Hoehn and Yahr tage, off	0.60	0.34	1.06	0.08	-	-	-	-
HAMA score	0.34	0.15	0.78	0.01	-	-	-	-
HAMD score	0.43	0.22	0.87	0.02	-	-	-	-
Limited peri-electrode edema on T2	4.76	1.09	20.8	0.04	7.63	1.37	42.37	0.02
Amantadine	2.38	0.94	6.04	0.07	3.64	1.08	12.28	0.04
Preoperative hypnotic	0.29	0.11	0.78	0.02	-	-	-	-

Significance was recognized when *p* < 0.05. PD: Parkinson’s disease, STN: subthalamic nucleus, DBS: deep brain stimulation, HAMA: Hamilton anxiety scale, HAMD: Hamilton depression scale.

## Data Availability

The research data are available upon request.

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
