# Peer review of "Risk Factors for Hiccups after Deep Brain Stimulation of Subthalamic Nucleus for Parkinson’s Disease"

_brainsci, 2022, doi:10.3390/brainsci12111447_

Round 1

Reviewer 1 Report

The authors reported an interesting study about risk factors for hiccup in PD patients with DBS. I have some comments to the authors:

-       English should be checked by a native speaker

-       Please better define timing of hiccup in your PD patients

Reviewer 2 Report

The authors investigated and found the risk factors for hiccups after deep brain stimulation of the STN for Parkinson’s disease. The potentially involved risk factors were analysed using multiple statistical tools, and satisfactory results were reported. A very well-developed literature review, including the applied methodology. Clear description of the surgical procedure, and the ways on how the data were collected.  

Despite the dedication shown by the authors when dealing with a proper scientific approach to their research, I am having some questions that need to be clarified/addressed as follows: 

1.       There is no mention of randomization when performing the research design. Please clarify. 

2.       Line 152 mentions several statistical techniques of very different nature, ranging from parametric and non-parametric tests, continuous and dichotomous variables, and association vs relationships tests. However, there is neither mention throughout the paper under which circumstances these tests were used, nor exposition of corresponding results/conclusions achieved.

3.       Following line 153, it seems that the authors tried to reduce the dimension of the data by using some univariate analysis. Please, explicitly specify which test/technique did you use for this task. In addition, could you please argue as to why unsupervised learning techniques were not used for choosing the suspected factors.

4.       Line 156, proper reference needed for the assumed standard of p<0.1.

5.       Line 169, authors argue to have used univariate logistic regression analysis to assess the difference between delirium and non-delirium group. Please clarify as this would imply running a model for every single variable separately, which is statistically unsound.

6.       Line 191. Similar issue as previous point. Also, which type of univariate and multivariate tests did you run?

7.       Line 206 describes a binary multivariable logistic regression (I assume the output to be: It did have postoperative hiccups or did not). Confidence intervals for the odd ratios look very wide that may render the results useless. Could the authors argue about this?

Reviewer 3 Report

The Manuscript: „ Risk factors for hiccups after deep brain stimulation of subthalamic nucleus for Parkinson’s disease’’ by Bin Wu and colleagues report for the first time the risk factors of hiccups after subthalamic nucleus deep brain stimulation under general anaesthesia for PD. Based on the results of the study, the authors identified that masculinity, higher BMI, preoperative use of amantadine and postoperative limited non-infectious peri-electrode edema in deep white matter being independent risk factors for postoperative hiccups. The study is nicely conducted with elaborate description of methodology and documentation of subsequent result. After going through the manuscript, I have following comments to the author:

1.        In the study only 34 patients had hiccups in comparison to 157 non-hiccup patients. Was a sample size analysis performed? If yes, was the sample size large enough to draw the statistical conclusions?

2.       How were the hiccup data of the patients extracted? From the clinical reports of the patients or the patients reported of hiccups themselves?
